# Citrullinated and Malondialdehyde–Acetaldehyde-Modified Fibrinogen Activates Macrophages and Promotes Coronary Endothelial Cell Inflammation

**DOI:** 10.3390/cimb47110943

**Published:** 2025-11-13

**Authors:** Wenxian Zhou, Hannah J. Johnson, Michael J. Duryee, Nozima Aripova, Engle E. Sharp, Carlos D. Hunter, Kimberley Sinanan, Henry C. Drvol, Mason G. Feely, Tate M. Johnson, Mabruka Alfaidi, Daniel R. Anderson, Vineeth K. Reddy, Keshore Bidasee, Robert G. Bennett, Jill A. Poole, Geoffrey M. Thiele, Ted R. Mikuls

**Affiliations:** 1Department of Internal Medicine, Division of Rheumatology, University of Nebraska Medical Center, Omaha, NE 68198, USA; wezhou@unmc.edu (W.Z.); hannjohnson@unmc.edu (H.J.J.); mduryee@unmc.edu (M.J.D.); nozima.aripova@unmc.edu (N.A.); esharp@unmc.edu (E.E.S.); cdhunter@unmc.edu (C.D.H.); ksinanan@unmc.edu (K.S.); hdrvol@unmc.edu (H.C.D.); mafeely@unmc.edu (M.G.F.); tate.johnson@unmc.edu (T.M.J.); gthiele@unmc.edu (G.M.T.); 2Veterans Affairs Nebraska-Western Iowa Health Care System, Omaha, NE 68105, USA; rgbennet@unmc.edu; 3Department of Cellular and Integrative Physiology, University of Nebraska Medical Center, Omaha, NE 68198, USA; malfaidi@unmc.edu; 4Department of Internal Medicine, Division of Cardiovascular Medicine, University of Nebraska Medical Center, Omaha, NE 68198, USA; danderso@unmc.edu (D.R.A.); vreddy@unmc.edu (V.K.R.); 5Department of Pharmacology & Experimental Neuroscience, University of Nebraska Medical Center, Omaha, NE 68198, USA; kbidasee@unmc.edu; 6Department of Internal Medicine, Division of Diabetes, Endocrinology and Metabolism, University of Nebraska Medical Center, Omaha, NE 68198, USA; 7Department of Biochemistry and Molecular Biology, University of Nebraska Medical Center, Omaha, NE 68198, USA; 8Department of Internal Medicine, Division of Allergy and Immunology, University of Nebraska Medical Center, Omaha, NE 68198, USA; japoole@unmc.edu

**Keywords:** malondialdehyde–acetaldehyde, citrulline, post-translational modifications, rheumatoid arthritis, heart failure, endothelial cell

## Abstract

Individuals with rheumatoid arthritis (RA) face increased cardiovascular mortality due to heart failure (HF) complications. Post-translational modifications, such as citrullination (CIT) and malondialdehyde–acetaldehyde (MAA) adduction, are implicated in RA pathogenesis. However, their role in RA-associated HF is not well understood. This study examines the deposition of MAA and CIT in cardiac tissues of RA-HF patients and investigates how MAA and CIT adducts on fibrinogen (FIB-MAA-CIT) drive crosstalk between macrophages and endothelial cells in vitro. We demonstrated elevated MAA and CIT adducts, strong perivascular MAA-CIT co-localization, and increased perivascular collagen deposition in the myocardium of RA-HF patients compared to non-RA HF controls. Treating human coronary artery endothelial cells (HCAECs) with FIB-MAA-CIT induced upregulation of inflammatory markers including MCP-1, IL-6, ICAM-1, and VCAM-1 compared to unmodified FIB. This response was amplified when HCAECs were treated with cell culture media obtained from FIB-MAA-CIT-stimulated macrophages. FIB-MAA-CIT activation of macrophages engaged NF-κB and p38 signaling pathways and inhibition of these pathways reduced FIB-MAA-CIT-mediated macrophage cytokine secretion and subsequent HCAEC responses. In summary, our findings support a novel mechanism by which endogenously modified proteins drive macrophage–endothelial cell crosstalk, promoting myocardial inflammation. Targeting these post-translational modifications may present novel therapeutic strategies to mitigate HF in RA.

## 1. Introduction

Rheumatoid arthritis (RA) is a systemic autoimmune disease affecting approximately 0.5–1% of the global population [1,2,3]. Individuals with RA have a 50% higher risk of cardiovascular disease-related mortality compared to the general population, with heart failure (HF) representing one of the most frequent and deadly complications [4,5,6,7]. Multiple mechanisms have been proposed for this increased cardiovascular burden, including systemic inflammation, oxidative stress, endothelial dysfunction, and post-translational modifications (PTMs) [4,8,9,10]. Macrophages and endothelial cells have been identified as important cell types in promoting the development of HF, particularly in the context of chronic inflammatory diseases such as RA, wherein systemic inflammation drives endothelial dysfunction leading to fibrosis, ischemia, and diastolic dysfunction [11,12]. However, the specific molecular mechanisms driving RA-related HF remain poorly defined.

PTMs have been extensively implicated in RA pathogenesis, promoting tolerance loss, autoantibody generation, inflammation, and fibrosis [13,14,15]. Citrullination (CIT), a PTM catalyzed by peptidyl arginine deiminase (PAD), is one of the most well-characterized modifications in RA [16]. CIT generates immune-reactive neoepitopes that are targeted by anti-citrullinated protein antibodies (ACPAs), which are highly specific to RA [17]. ACPAs target citrullinated fibrinogen [18], vimentin [19], and type II collagen [20], among many other self-proteins, promoting joint damage and extra-articular complications. Anti-cyclic citrullinated peptide (CCP) antibodies, a commercial measure of ACPAs, have been associated with impaired diastolic and systolic function and increased left ventricular mass [21,22,23,24], suggesting their potential role in promoting cardiac remodeling that leads to HF. In addition, increased expression of myocardial citrulline and reduced calcium sensitivity have been linked to reduced cardiomyocyte contractility in mice during the early stages of collagen induced arthritis, suggesting that citrullinated proteins may contribute to the early development of cardiac dysfunction in RA [25]. Moreover, citrullinated proteins are significantly increased in myocardial tissues of RA patients compared to controls, further supporting their potential role in promoting cardiovascular complications in RA [26].

Malondialdehyde–acetaldehyde (MAA), another PTM structurally distinct from CIT, represents an alternative autoantibody target in RA [27]. MAA is a byproduct of lipid peroxidation and induces robust autoantibody responses even in the absence of adjuvant [28]. Elevated concentrations of circulating anti-MAA antibodies targeting modified fibrinogen, albumin, vimentin, and collagen have been reported in RA and associated with other fibrotic extra-articular manifestations, including RA interstitial lung disease [27,29,30]. Additionally, IgA anti-MAA antibodies have been associated with increased coronary artery calcium, insulin resistance, and decreased high-density lipoprotein particle number, in addition to improving the 10-year prediction of coronary atherosclerosis in RA beyond that afforded by the standard cardiovascular risk calculation [31].

In addition to elevated autoantibody levels, MAA adducts strongly co-localize with citrullinated proteins and are enriched in the joints and lungs of RA patients [27,29]. This co-localization is relevant, as recent studies have demonstrated that fibrinogen co-modified with both MAA and CIT synergistically activate macrophages, promoting inflammation and the subsequent activation of joint and lung fibroblasts to initiate fibrotic responses in vitro [23,24,25]. Moreover, MAA adducts have been shown to activate vascular endothelial cells [28], while CIT has been observed to activate retinal endothelial cells [32]. Despite increasing evidence implicating these PTMs in RA-related cardiovascular disease, no prior studies have examined whether MAA is present and co-localizes with CIT in myocardial tissues, or whether these modifications, alone or in combination, generate pro-inflammatory effects in cardiac cell types.

To address this evidence gap, we tested the hypotheses that MAA and CIT are enriched and co-localized in the myocardium of individuals with RA and concomitant HF and that fibrinogen co-modified with MAA and CIT synergistically increase inflammation in human coronary artery endothelial cells (HCAECs). In addition, we propose that macrophages stimulated with co-modified fibrinogen release soluble mediators that exacerbate the release of inflammatory mediators by HCAECs, potentially leading to perivascular fibrosis.

## 2. Materials and Methods

### 2.1. Study Subjects

To evaluate whether MAA and CIT are enriched and co-localized in myocardial tissues of individuals with RA, left ventricular (LV) apex tissues were obtained from patients diagnosed with both RA and advanced heart failure (RA-HF; n = 3) and accessed from the Nebraska Cardiovascular Biobank (Appendix A) [33]. Fresh LV apex tissues were collected either from punch biopsy of patients undergoing LV assist device (LVAD) placement or the dissection of LV apex tissues from native hearts removed at the time of transplant surgery (Appendix A). As only 3 heart tissues were available from patients with RA-HF at the time of this study, no other formal inclusion/exclusion criteria were applied. LV apex tissues were also obtained from age- and sex-matched non-RA controls with advanced HF (n = 3) (Appendix A). All tissues were derived from donors with reduced LV ejection fraction at the time of procurement. Information specific to the underlying etiology of cardiac dysfunction was not available. Informed consent was obtained from all tissue donors under the University of Nebraska Medical Center (UNMC) IRB protocol No. 0099-24-EP. Heart tissues were preserved in Allprotect Tissue Reagent (Qiagen, Hilden, Germany) and stored at −80 °C until use.

To validate results from a U-937 monocytic cell line, peripheral blood was collected from five healthy donors (Appendix A) and used immediately for subsequent experiments. Both male and female donors were included to enhance the generalizability of our results. Healthy donors provided informed consent under UNMC IRB protocol No. 608-22-EP.

### 2.2. Immunohistochemistry of Myocardial Tissues

LV apex tissues from 3 RA-HF and 3 non-RA HF donors were thawed, fixed in 4% paraformaldehyde for 24 h, dehydrated in 70% ethanol for 24 h, embedded in paraffin, sectioned into 4–5 μm slices, and placed on glass slides. These tissues then underwent paraffin removal with xylene for 5 min, followed by sequential rehydration in 100% ethanol, 95% ethanol, 80% ethanol, 50% ethanol, deionized water, and PBST for 1 min each. One section per donor was stained with Masson’s trichrome to detect collagen deposition indicative of tissue fibrosis. Another section from each donor underwent immunohistochemistry to stain for MAA and CIT deposition. These sections were subjected to antigen retrieval using 10 mM citrate buffer (pH 6.0) for 40 min using a steamer. After blocking with 2% goat serum, sections were incubated overnight at 4 °C with primary antibodies and isotype controls (1:100 dilution). Primary antibodies included Zenon AF 594-labeled (Invitrogen, Carlsbad, CA, USA) rabbit polyclonal anti-MAA IgG and mouse monoclonal anti-peptidyl-CIT antibody (Millipore Sigma, Burlington, MA, USA; clone F95). AF647 Rabbit IgG and AF 488 Mouse IgM (1:100 dilution) were used as isotype controls for anti-MAA and anti-CIT antibodies, respectively. Detection of anti-CIT antibody was carried out using AF488 AffiniPure Donkey anti-mouse IgM (1:1000; Jackson ImmunoResearch, West Grove, PA, USA). After a 45 min incubation with secondary antibody at room temperature, 4′,6-diamidino-2-phenylindole (DAPI) was applied, and samples were mounted with Fluoromount-G (Southern Biotech, Birmingham, AL, USA). Imaging was performed on a Nikon Eclipse inverted microscope (Nikon, Melville, NY, USA). Five representative images were taken and averaged for each tissue.

Immunohistochemistry quantification and co-localization analysis were conducted as previously described using ImageJ (version 2.14.0) and the FIJI Coloc2 plugin (version 3.0.5) (NIH, Bethesda, MD, USA) [27,29]. Briefly, the mean pixel density of CIT (green) and MAA (red) antigens was quantified by the “measure” function. Pearson’s R values (with no threshold) were generated as measures of co-localization. Quantification of perivascular and interstitial fibrosis has also been described in previous publications [34,35]. Briefly, five representative images were taken of tissue from each donor at 20× magnification at both perivascular regions (regions containing blood vessels) and interstitial regions (regions not containing blood vessels). Mean intensity of collagen deposition (blue) in each image was quantified using the “colour deconvolution” and “measure” functions in FIJI.

### 2.3. MAA and CIT Antigen Preparation

Fibrinogen (FIB; Cayman Chemical, Ann Arbor, MI, USA) was selected for modification due to its established role as an ACPA and anti-MAA autoantibody target and its known capacity to activate macrophages following post-translational modification [18,36,37,38]. FIB antigens were prepared as previously described [36,37,38].

For MAA modification (FIB-MAA), FIB was incubated with 8.0 mM malondialdehyde (Aldrich Chemical Co., Milwaukee, WI, USA) and 4.0 mM acetaldehyde (Aldrich Chemical Co.) in 0.1 M phosphate buffer (pH 7.2) at 37 °C for 3 days. Proteins were then dialyzed three times with a Spectrum^TM^ dialysis membrane (6–8 kDa cutoff; Thermo Fisher, Waltham, MA, USA) in 0.1 M phosphate buffer at 4 °C for 24 h. MAA modification was confirmed by measuring the fluorescence of the dihydropyridine ring structure (excitation 398 nm, emission 460 nm) using a spectrofluorometer (Turner Biosystems, Sunnyvale, CA, USA).

For citrullination (FIB-CIT), FIB was incubated with rabbit skeletal PAD-4 enzyme (4 unit/mg protein) (Cayman Chemical) at 37 °C for 2 h. Excess PAD-4 was then removed with an excess amount of immobilized soybean trypsin inhibitor (Thermo Fisher) [39]. Citrullination was confirmed by ELISA and Western blot.

For dual modification (FIB-MAA-CIT), MAA modification was performed prior to citrullination based on prior studies showing enhanced macrophage responses with this order [40]. All modified FIB preparations were sterile-filtered (0.2 μm) and confirmed to have undetectable endotoxin levels by the Limulus Amebocyte Lysate Assay (Lonza, Bend, OR, USA).

### 2.4. Human Macrophage Culture, Antigen Stimulation, and Supernatant Collection

U-937 cells (RRID: CVCL0007), a human monocytic cell line derived from a 37-year-old White male, were purchased from ATCC (Manassas, VA, USA). The commercial supplier completed mycobacterium testing before shipment. Cells were cultured in RPMI medium (Thermo Fisher) supplemented with 10% fetal bovine serum (FBS), 2 mM L-glutamine, penicillin–streptomycin, and 0.05 mM 2-mercaptoethanol at 37 °C in a 5% CO_2_ incubator. At passage 5–7, U-937 cells were differentiated into macrophages (MϕU) by treatment with 100 ng/mL phorbol 12-myristate 13-acetate (PMA; Sigma-Aldrich, St. Louis, MO, USA) overnight, followed by a 48 h resting period with daily media changes. MϕU were then stimulated with 25 μg/mL of unmodified FIB, FIB-MAA, FIB-CIT, or FIB-MAA-CIT (Appendix A).

Peripheral blood mononuclear cells (PBMCs) from five healthy donors (Appendix A) were isolated using SepMate tubes following the manufacturer’s instructions (StemCell Technologies, Vancouver, BC, Canada). Immediately following isolation, PBMCs were cultured in RPMI 1640 supplemented with 10% FBS, 2 mM L-glutamine, penicillin–streptomycin, and 0.05 mM 2-mercaptoethanol. Subsequently, PBMCs were differentiated into M0 macrophages (MϕP) by stimulating with 100 ng/mL of macrophage colony-stimulating factor (M-CSF; Thermo Fisher) for 7 days. MϕP were then stimulated with 25 μg/mL of unmodified FIB, FIB-MAA, FIB-CIT, or FIB-MAA-CIT (Appendix A).

### 2.5. Coronary Endothelial Cell Culture & Stimulation

HCAECs were purchased from Cell Applications, Inc. (San Diego, CA, USA). These cells are sourced from the human coronary artery of a 19-year-old Hispanic male. HCAECs were grown in human MesoEndo growth medium (Cell Applications) and seeded to 24-well plates between passage 5–7. After achieving 70% confluency, HCAECs were stimulated with standard volumes (25% of total well volume) of MϕU supernatant (MϕU-SN) or MϕP supernatant (MϕP-SN) collected 48 h post-treatment with modified FIB antigens, as detailed above. To account for the potential influence of residual FIB antigens in MϕU-SN or MϕP-SN, HCAECs were also directly stimulated with 25 μg/mL of unmodified FIB, FIB-MAA, FIB-CIT, or FIB-MAA-CIT for comparison in separate experiments (Appendix A). HCAECs were stimulated with media as a negative control and demonstrated negligible evidence of activation.

### 2.6. HCAEC Gene Expression and Cytokine Secretion

A gene expression panel was utilized to explore the effect of MϕU-SN on HCAECs. Briefly, HCAECs were stimulated with MϕU-SN for 8 h. RNA was extracted from HCAECs per the manufacturer’s protocol using RNeasy Mini Kit (Qiagen) and the concentration quantified using the Nanodrop 2000c (Thermo Fisher). Total RNA was hybridized and processed as per manufacturer’s protocol using the nCounter^®^ Cardiovascular Disease Pathophysiology Panel (800 genes) (NanoString, Seattle, WA, USA). Resulting data were normalized, and grouped, and differential expression was calculated using nSolver advanced analysis software (version 4.0) (NanoString). The Benjamini–Yekutieli procedure was used to calculate an adjusted *p*-value to reduce the false discovery rate. Pathway analysis was performed using QIAGEN Ingenuity Pathway Analysis software (version 2025.2).

To validate NanoString findings and explore the temporal aspect of gene expression, HCAECs were stimulated with modified FIB or MϕU-SN for either 8 h, 1 day, or 7 days. For 7-day stimulations, endothelial cell media were exchanged every 48 h (days 2, 4, and 6) with fresh unmodified/modified FIB or MϕU-SN generated under identical conditions. After stimulation, HCAEC supernatants were collected and stored at −80 °C until analysis. For 7-day stimulations, supernatants from the last stimulation (day 6–7) were collected and stored. At the end of each stimulation time point, RNA was extracted and quantified and then converted to cDNA using a High-Capacity RNA-to-cDNA kit (Applied Biosystems, Waltham, MA, USA). To make the gene amplification mixture, cDNA was mixed with TaqMan Gene Expression Master Mix (Applied Biosystems) and TaqMan Gene Expression Assays (Thermo Fisher). The following TaqMan primer/probe sets were used in real-time polymerase chain reaction (RT-PCR) analysis: *CCL2 (MCP1)*, *IL6*, *Vascular Cell Adhesion Molecule 1 (VCAM1)*, and *Intercellular Adhesion Molecule-1 (ICAM1)* (Thermo Fisher). These four genes were selected as they represented the top upregulated genes identified by our RNA NanoString profiling regarding inflammatory cytokines and leukocyte adhesion molecules, respectively. These genes are also important promotors of endothelial dysfunction and heart failure [12,41,42,43,44]. Gene expression was normalized to *GAPDH*. Normalized gene expressions were compared to the medium control and presented as relative quantification (RQ).

To test whether gene expression paralleled protein secretion, concentrations of soluble proteins in HCAEC supernatants were measured by ELISA. The following ELISAs were performed per the manufacturer’s protocol (Biolegend, San Diego, CA, USA) and analyzed on an Epoch plate reader using Gen 5 software: MCP-1, IL-6, ICAM-1, and VCAM-1. For the 7-day experiment, protein secretion during the last stimulation (day 6–7) were measured. To account for background cytokines present in MϕU-SN or MϕP-SN, macrophage cytokine concentrations were subtracted from the corresponding HCAEC supernatant values, with adjusted HCAEC supernatant protein concentrations reported.

### 2.7. Macrophage Intracellular Signaling

To delineate intracellular signaling mechanisms involved in macrophage activation, MϕU were lysed 1 h after antigen stimulation in radioimmunoprecipitation assay (RIPA) buffer (Thermo Fisher Scientific) in the presence of a protease inhibitor cocktail (Roche, Basel, Switzerland) and 1 mM phenylmethylsulfonyl fluoride (PMSF, Sigma Aldrich). Protein concentrations were quantified using the bicinchoninic acid (BCA) assay (Thermo Fisher) and equated between samples by diluting in deionized water. Cellular lysates (25 μg) were then loaded onto 4–20% gradient gels and subjected to SDS-PAGE electrophoresis for Western blot (Bio-Rad Hercules, CA, USA). Gels were transferred to a polyvinylidene difluoride (PVDF) membrane (EMD Millipore Corp, Burlington, MA, USA), blocked in 2% casein (Sigma-Aldrich), and incubated with primary antibodies (1:1000) overnight. Primary rabbit antibodies used are listed in the Major Resources Table.

The following day, blots were incubated with horseradish peroxidase (HRP)-conjugated goat anti-rabbit IgG (1:10,000) (Jackson ImmunoResearch). For blot development, West Pico Chemiluminescent Substrate (Thermo Fisher) was used. Blots were visualized using the KwikQuant imager (Kindle Biosciences LLC, Greenwich, CT, USA). Respective band intensities were measured using KwikQuant Image Analyzer software (v. 1.8.6) and normalized to β-actin. After the detection of phosphorylated proteins, blots were incubated with stripping solution (25 mM glycine-HCl, 1% SDS, PH 2.0) for 30 min, then washed, blocked, and stained for total proteins as described above. The ratio between phosphorylated and total proteins were calculated to quantify signaling pathway activation.

The following signaling pathways were evaluated: phosphorylation of p38 mitogen-activated protein kinase (p38), nuclear factor kappa B (NF-κB), extracellular signal-regulated kinases 1/2 (Erk1/2), stress-activated protein kinases (SAPK)/Jun amino-terminal kinases (JNK), AKT serine/threonine kinase (AKT)1, AKT2, signal transducer and activator of transcription (STAT)1, STAT3, and STAT6. These signal transduction pathways were selected for investigation as each has been implicated in the polarization and function of macrophages [45,46]. The 1 h stimulation time point was selected because it was previously reported in our laboratory to produce optimal phosphorylation of intracellular signaling molecules [37].

### 2.8. Inhibition of P38 and NF-kB Signaling in MϕU or MϕP

To test whether p38 and NF-κB activation are functionally required for FIB-MAA-CIT-induced cytokine secretion and to dissect pathway specificity, inhibitors of these pathways were utilized. MϕU and MϕP were pre-incubated with 1 µM of BIRB-796 (p38 inhibitor; Cayman Chemical) and/or 50 µM of BAY-11-7085 (NFkB inhibitor; Cayman Chemical) for 1 h before stimulation with unmodified/modified FIB antigens. These inhibitor concentrations were selected as they represent the minimum concentrations required to inhibit the respective intracellular signaling pathways in vitro [47,48]. MϕU and MϕP were then stimulated with FIB antigens in combination with each inhibitor. One hour after antigen stimulation, protein lysates were collected and Western blot was performed as described above to confirm complete inhibition of the respective signaling pathways in macrophages. Two days after antigen stimulation, supernatants were collected and used to stimulate HCAECs, as described above. Prior to stimulating HCAECs, concentrations of tumor necrosis factor-α (TNF-α), interleukin (IL)-1β, IL-6, and monocyte chemoattractant protein 1 (MCP-1) in MϕU-SN and MϕP-SN were measured by ELISA.

### 2.9. Statistical Analyses

Comparisons of antigen and collagen deposition between RA-HF and Non-RA HF tissues in Figure 1 were analyzed using Student’s *t*-test, after confirming normality (*p* > 0.05) using the Shapiro–Wilk test. Differential gene expression in Figure 2 was analyzed using the Benjamini–Yekutieli procedure to reduce the false discovery rate. Comparisons between different modified antigens and between direct and indirect stimulation in Figure 3 and Figure 4 were analyzed using two-way ANOVA with Tukey’s post hoc test (95% confidence interval) to account for multiple comparisons. Comparisons among different modified antigens in Figure 5, Figure 6, Figure 7 and Figure 8 were analyzed using one-way ANOVA with Tukey’s post hoc test (95% confidence interval). Statistical significance was defined as *p* < 0.05 after multiple comparison adjustments. Graphs display mean ± standard deviation. The Benjamini–Yekutieli procedure was performed using Nsolver advanced analysis software (NanoString). Remaining analyses were performed using GraphPad Prism 10.4.2 (San Diego, CA, USA).

## 3. Results

### 3.1. Increased MAA and CIT Expression, Strong MAA-CIT Co-Localization, and Increased Perivascular Fibrosis in RA-HF Myocardium

Immunohistochemistry revealed significantly higher expression of MAA (*p* < 0.01) and CIT adducts (*p* < 0.05) in LV apex tissues of RA-HF patients compared to non-RA HF controls (Figure 1A and Appendix A). MAA and CIT adducts were expressed in both the myocardium and around perivascular spaces. Notably, strong co-localization (r^2^ > 0.7) between MAA and CIT adducts was observed in both HF groups (Figure 1A). Negative controls using isotype control antibodies confirmed minimal background staining (Appendix A). Additionally, myocardial tissues from individuals with RA-HF exhibited significantly increased perivascular collagen deposition (*p* < 0.05) indicative of fibrosis (Figure 1B and Appendix A). RA-HF myocardium demonstrated similar differences compared to non-RA HF tissues in interstitial collagen deposition, though this difference did not achieve statistical significance (*p* = 0.23).

### 3.2. FIB-MAA-CIT Promotes Coronary Endothelial Inflammation In Vitro via Macrophage Interaction

Because MAA and CIT adducts were observed in proximity to areas of perivascular fibrosis of RA-HF tissues (Figure 1), we explored whether supernatants from FIB-MAA-CIT-stimulated macrophages promote differential gene expression suggestive of increased HCAEC activation. HCAECs were indirectly stimulated with supernatants from U-937-derived macrophages stimulated with unmodified and modified forms of FIB (MϕU-SN^FIB^, MϕU-SN^FIB-MAA^, MϕU-SN^FIB-CIT^, or MϕU-SN^FIB-MAA-CIT^) for 8 h. MϕU-SN^FIB-MAA-CIT^ induced significant differential gene expression suggestive of greater endothelial cell activation compared to all other treatment groups (Figure 2A and Appendix A). Out of 800 genes included on the cardiovascular panel, MϕU-SN^FIB-MAA-CIT^ induced significant upregulation of 31 genes and downregulation of 38 genes compared to MϕU-SN^FIB^ (Appendix A). The top 30 differentially expressed genes are shown in Figure 2B, with numbers indicating log2 fold change versus MϕU-SN^FIB^. HCAECs stimulated with MϕU-SN^FIB-MAA-CIT^ exhibited a marked increase in endothelial inflammation, as evidenced by upregulated gene expression of *CCL2*, *NFKBIA*, *IL6*, *CXCL8*, *PTGS2*, *VCAM1*, and *ICAM1*, along with downregulation of *PRKACB*. These cells also showed decreased expression of endothelial markers (*PECAM1* and *VWF*) and increased expression of angiogenesis-associated genes such as *HDGF* and *HIF1A* (Figure 2B).

Results of pathway enrichment analyses are shown in Figure 2C,D. MϕU-SN^FIB-MAA-CIT-^ stimulated HCAECs exhibited upregulation of tumor microenvironment pathways, IL-4 and IL-13 signaling, inflammation signaling, and fibrosis signaling, with concurrent downregulation of IL-10 signaling pathways (Figure 2D). Overall, HCAEC responses following MϕU-SN^FIB-MAA-CIT^ stimulation are predominantly inflammatory at 8 h, with additional evidence of mesenchymal transition, as indicated by elevated tumor microenvironment and fibrosis signaling.

To validate the NanoString results, gene expression and protein secretion of MCP-1, IL-6, ICAM-1, and VCAM-1 were measured 8 h post-stimulation using RT-PCR (Figure 3A–D) and ELISA (Figure 4A–D), respectively. Direct stimulation with FIB-MAA-CIT resulted in the highest increase in gene expression of *MCP1* (*p* < 0.01 vs. FIB), *IL6* (*p* < 0.01 vs. FIB), *ICAM1* (*p* < 0.01 vs. FIB), and *VCAM1* (*p* < 0.001 vs. FIB) compared to direct stimulation with single- or unmodified FIB (Figure 3A–D, gray bars). However, FIB-MAA-CIT direct stimulation only significantly increased protein secretion of MCP-1 (*p* < 0.01 vs. FIB) and VCAM-1 (*p* < 0.05 vs. FIB), with no significant effects on IL-6 and ICAM-1 levels (Figure 4A–D, gray bar). On the other hand, indirect stimulation with MϕU-SN^FIB-MAA-CIT^ led to the highest upregulation in gene expression of *MCP1* (*p* < 0.0001 vs. MϕU-SN^FIB^), *IL6* (*p* < 0.0001 vs. MϕU-SN^FIB^), *ICAM1* (*p* < 0.01 vs. MϕU-SN^FIB^), and *VCAM1* (*p* < 0.001 vs. MϕU-SN^FIB^) compared to all other indirect stimulation groups (Figure 3A–D, pink bar). In addition, MϕU-SN^FIB-MAA-CIT^ indirect stimulation significantly increased protein secretion of MCP-1 (*p* < 0.01 vs. MϕU-SN^FIB^), IL-6 (*p* < 0.0001 vs. MϕU-SN^FIB^), and VCAM-1 (*p* < 0.0001 vs. MϕU-SN^FIB^), but not ICAM-1 (Figure 4A–D, pink bar). Importantly, the effects of indirect stimulation through macrophage supernatants were markedly stronger than direct antigen stimulation in terms of both endothelial gene expression and protein secretion (*p* < 0.0001, MϕU-SN^FIB-MAA-CIT^ vs. FIB-MAA-CIT) (Figure 3 and Figure 4).

To assess the durability of these responses, additional measurements were performed 24 h after stimulation. At 24 h, FIB-MAA-CIT direct stimulation continued to induce the highest increase in gene expression of *MCP1* (*p* < 0.0001 vs. FIB), *IL6* (*p* < 0.001 vs. FIB), *ICAM1* (*p* < 0.01 vs. FIB), and *VCAM1* (*p* < 0.05 vs. FIB) compared to direct stimulation with single- or unmodified FIB (Figure 3E–H, gray bar). However, FIB-MAA-CIT direct stimulation only caused a significant increase in protein secretion of ICAM-1 (*p* < 0.05 vs. FIB) and VCAM-1 (*p* < 0.05 vs. FIB), with no effect on the protein secretion of MCP-1 or IL-6 (Figure 4E–H, gray bar). Indirect stimulation with MϕU-SN^FIB-MAA-CIT^ resulted in the highest upregulation in gene expression of *MCP1* (*p* < 0.0001 vs. MϕU-SN^FIB^), *IL6* (*p* < 0.0001 vs. MϕU-SN^FIB^), *ICAM1* (*p* < 0.0001 vs. MϕU-SN^FIB^), and *VCAM1* (*p* < 0.0001 vs. MϕU-SN^FIB^) compared to all other indirect stimulation groups at 24 h (Figure 3E–H, pink bar). Similarly, MϕU-SN^FIB-MAA-CIT^ indirect stimulation caused increased protein secretion of all four markers (*p* < 0.0001 vs. MϕU-SN^FIB^) (Figure 4E–H, pink bar). Notably, MϕU-SN^FIB^ stimulation caused a decrease in VCAM-1 secretion to below the endothelial media stimulated baseline, but MϕU-SN^FIB-MAA-CIT^ increased VCAM-1 secretion back to above the baseline (Figure 4H).

Comparing the two time-points following MϕU-SN^FIB-MAA-CIT^ stimulation, gene expression of adhesion molecules (*ICAM1* and *VCAM1*) peaked early (at 8 h), whereas gene expression of cytokines (*MCP1* and *IL6*) peaked later (at 24 h), with approximately 2- to 5-fold elevations compared to corresponding levels after 8 h. (Figure 3). Protein secretion of all four inflammatory markers (MCP-1, IL-6, ICAM-1, and VCAM-1) peaked at 24 h (Figure 4). Compared to 24 h stimulation, the 7-day stimulation demonstrated similar gene expression and protein secretion patterns but with a reduced magnitude of response (Appendix A).

### 3.3. FIB-MAA-CIT Induces NF-κB and p38 Pathway Activation in Macrophage

Since FIB-MAA-CIT-mediated HCAEC activation appeared to be largely driven through macrophage interactions, we investigated the intracellular signaling mechanisms responsible for MϕU activation. After 1 h of stimulation, NF-κB, p38, Erk1/2, STAT1, STAT3, STAT6, Akt1, Akt2, and SAPK/JNK pathways in MϕU were screened using Western blot. FIB-MAA-CIT stimulation of MϕU resulted in significantly higher phosphorylation of NF-κB compared to all other groups (*p* < 0.01). Similarly, FIB-MAA-CIT stimulation led to the highest p38 phosphorylation (*p* < 0.05 vs. FIB and FIB-MAA; non-significant vs. FIB-CIT) (Figure 5 and Appendix A). FIB-MAA-CIT stimulation did not affect the phosphorylation of Erk1/2 or STAT1 (Figure 5). Phosphorylated SAPK/JNK, Akt1, Akt2, STAT3, and STAT6 were undetectable under all antigen stimulation conditions (Appendix A).

As NF-κB and p38 were upregulated in MϕU by FIB-MAA-CIT, we tested the effect of inhibiting these pathways. NF-κB inhibitor (BAY-11-7085) and p38 inhibitor (BIRB-796) completely suppressed phosphorylated NF-κB and p38, respectively, from FIB-MAA-CIT-stimulated MϕU, and the inhibitors demonstrated minimal cross-reactivity (Appendix A). ELISA assays showed that each inhibitor significantly reduced FIB-MAA-CIT-mediated secretion of inflammatory cytokines (MCP-1, IL-6, TNF-α, and IL-1β) from both MϕU (Figure 6A–D) and MϕP (Figure 6E–H). BAY-11-7085 completely suppressed all cytokine production to or below baseline. BIRB-96 completely reduced TNF-α (Figure 6C,G) and IL-1β (Figure 6D,H) secretion to baseline but only partially reduced MCP-1 (Figure 6A,E) and IL-6 (Figure 6B,F) secretion.

### 3.4. NF-kB and p38 Inhibition in Macrophages Attenuates Mϕ-SN^FIB-MAA-CIT^-Mediated HCAEC Inflammation

Incubating MϕU with either NF-κB or p38 inhibitors during FIB-MAA-CIT stimulation significantly reduced the MϕU-SN^FIB-MAA-CIT^-mediated upregulation of MCP-1, IL-6, ICAM-1, and VCAM-1 in HCAECs at both mRNA and protein levels (*p* < 0.0001). Both NF-κB and p38 inhibitors reduced the gene expression of *MCP1, ICAM1*, and *VCAM1* to the baseline levels with similar efficacy (Figure 7A,B,D), while the NF-κB inhibitor was more effective in reducing *IL6* gene expression in MϕU-SN^FIB-MAA-CIT^-stimulated HCAECs (Figure 7B). For protein secretion, both inhibitors reduced the secretion of VCAM-1 to below the baseline with similar efficacy (Figure 8D). The NF-κB inhibitor was more efficacious than the p38 inhibitor in reducing MCP-1, IL-6, and ICAM-1 secretion from MϕU-SN^FIB-MAA-CIT^-stimulated HCAECs (*p* < 0.0001, Figure 8A–C). These findings were confirmed using MϕP-SN^FIB-MAA-CIT^ (PBMC-derived macrophages from 5 separate human donors), showing similar responses (Figure 7E–H and Figure 8E–H).

## 4. Discussion

This study is the first to our knowledge to demonstrate elevated MAA adduct expression, strong MAA-CIT co-localization, and evidence of increased perivascular fibrosis in heart tissues of RA-HF patients compared to non-RA HF controls. In addition, this is the first report to examine FIB-MAA-CIT activation of any cardiac cell type and the first to reveal NF-κB and p38 as novel pathways activated in response to FIB-MAA-CIT stimulation of macrophages. We demonstrated herein that FIB-MAA-CIT promotes macrophage–endothelial cell crosstalk in vitro that culminates in the exacerbation of coronary endothelial cell inflammation. In addition to inflammation, our NanoString also revealed downregulation of *PECAM1/VWF* and upregulation of fibrotic pathways, supporting a novel hypothesis that MϕU-SN^FIB-MAA-CIT^-stimulated endothelial cells may begin to lose endothelial characteristics and start to acquire a pro-fibrotic, mesenchymal phenotype.

Modified fibrinogen was utilized in this study for cell stimulation because it is a well-known antigen targeted by both ACPA and anti-MAA antibodies and has been previously shown to activate macrophages in vitro following its post-translational modification with MAA and CIT [18,36,37,38]. Our study found that FIB-MAA-CIT promotes macrophage–endothelial crosstalk in vitro, increasing the secretion of IL-1β, TNF-α, ICAM-1, and VCAM-1. These results align with a prior histopathology study demonstrating elevated levels of the same inflammatory mediators in the perivascular regions of RA myocardium compared to patient-matched skeletal muscle and myocardial tissues of non-RA controls [49]. Our data supports a possible mechanism linking MAA and CIT modification to myocardial inflammation.

While fibrinogen modified with either MAA or CIT alone showed negligible to no effects on coronary endothelial cell activation, MAA-CIT co-modified fibrinogen activated HCAECs, resulting in significantly increased gene expression of *MCP1*, *IL6*, *ICAM1*, and *VCAM1,* along with modest increases in protein secretion of MCP-1, ICAM-1, and VCAM-1. Collectively, these results demonstrate a synergistic role of MAA and CIT adducts in promoting endothelial cell inflammation in vitro. Notably, these effects were further amplified through macrophage–endothelial crosstalk, whereby indirect stimulation using Mϕ-SN^FIB-MAA-CIT^ promoted significantly higher secretion of all four markers compared to direct antigen stimulation. Circulating concentrations of MCP-1, IL-6, ICAM-1, and VCAM-1 are known to be elevated in patients with RA [50,51,52]. In addition, these soluble mediators have each been shown to promote cardiac fibrosis and dysfunction in animal models [41,42,43,44]. Together, these findings demonstrate for the first time that these soluble mediators secreted by Mϕ-SN^FIB-MAA-CIT^-stimulated HCAECs may be important drivers of the perivascular fibrosis observed in RA-HF myocardium.

Our results also revealed that co-modified fibrinogen activates macrophages via NF-κB and p38 signaling pathways. We demonstrate herein that these pathways are crucial for both the FIB-MAA-CIT-mediated macrophage activation and subsequent macrophage–endothelial crosstalk in vitro. The NF-κB pathway is among the most common aberrantly expressed pathways in RA and is a key regulator of inflammation [53]. Methotrexate, a first-line therapy in RA, has been shown to inhibit both MAA adduct formation and the NF-κB pathway [54,55]. Notably, prior work has also demonstrated that methotrexate use in RA is associated with an approximate 60% reduction in HF-related hospitalization, an effect that was independent of treatment-related reductions in clinically assessed RA disease activity [56]. However, whether methotrexate improves RA-HF prognosis by inhibiting FIB-MAA-CIT-mediated macrophage–endothelial cell crosstalk remains unknown.

With questions remaining regarding the potential safety and efficacy of agents inhibiting NF-κB and p38 signaling [53,57,58,59,60,61], our findings suggest that MAA and/or CIT might serve as novel ‘upstream’ therapeutic targets for mitigating cardiovascular complications in RA. PAD inhibitors, which suppress CIT formation, have already demonstrated efficacy in reducing cardiac dysfunction in collagen-induced arthritis models [62]. Likewise, MAA formation can be attenuated through administration of reactive aldehyde species inhibitors [63,64], though these agents have yet to be systemically tested in RA or HF. Whether effective inhibition of MAA or CIT could reduce endothelial inflammation and improve cardiac outcomes in RA patients remains to be seen.

### Limitations

There are limitations to this study. The small number of cardiac tissues available for this study renders it sensitive to outliers and individual heterogeneity. In addition, the lack of adjudication of heart failure subtype and RA disease characteristics could limit the RA-specific inferences from these observations. Notably, all RA-HF patients and non-RA HF controls had advanced heart failure with ejection fractions below 35% at the time of tissue collection. Moreover, information specific to the underlying etiology of myocardial dysfunction was unknown. This limitation is relevant as patients with RA are at a particularly heightened risk for developing HF with preserved ejection fraction (HFpEF), a condition primarily driven by cardiac inflammation and fibrosis [5,6,65]. Whether MAA- and CIT- modified proteins are specifically enriched in heart tissues of individuals with RA-HFpEF, and whether these PTMs contribute to HF disease progression in RA, remains to be investigated. Additionally, cardiac tissues from RA-HF patients exhibited substantial fatty infiltration that was not apparent in the non-RA HF specimens, which could potentially confound measurements of tissue fibrosis [66]. However, increased perivascular collagen deposition in RA-HF tissues remained evident even after excluding regions of fatty infiltration. Although the limited number of samples examined renders our results susceptible to outliers and may limit generalizability to larger populations, these data are consistent with preliminary observations in collagen-induced arthritis mice in addition to prior human studies showing increased citrullination in RA myocardium and the presence of MAA in coronary atheromas [26,67,68].

The 25 µg/mL dose of modified fibrinogen was used for cell stimulations because it has been previous previously shown as the optimal concentration to activate macrophages in our in vitro model [36,37,38]. However, whether this concentration reflects the in vivo concentration of modified fibrinogen in myocardial tissue is unknown and warrants future investigation. In addition, the residual antigen carry-over in macrophage supernatants was not quantified. To account for the potential influence of residual antigens in macrophage supernatant, endothelial cells were also directly stimulated with modified antigens for comparison in separate experiments.

This study primarily focused on inflammatory endothelial cell responses, as this was the predominant response that we observed 8 h after indirect stimulation with MϕU-SN^FIB-MAA-CIT^. However, our NanoString data also showed downregulation of endothelial markers such as *PECAM1* and *VWF*, alongside upregulation of the tumor microenvironment and fibrosis-related pathways in HCAECs. These data support a novel hypothesis that MϕU-SN^FIB-MAA-CIT^-stimulated endothelial cells may begin to lose endothelial characteristics and start to acquire a pro-fibrotic, mesenchymal phenotype. Previous studies have shown that chronic stimulation of endothelial cells with IL-1β, TNF-α, or TGF-β can induce transformation to a mesenchymal phenotype [69,70,71]. Since our current and previous studies [36] have shown that co-modified fibrinogen increases macrophage secretion of IL-1β, TNF-α, and TGF-β, chronic indirect stimulation with Mϕ-SN^FIB-MAA-CIT^ could lead to a robust pro-fibrotic phenotype in endothelial cells, further contributing to perivascular fibrosis in RA. Future studies will be needed to characterize this pro-fibrotic endothelial phenotype more precisely and to identify the core mediators of this macrophage–endothelial crosstalk.

Our previous research has shown MAA and CIT co-modification leads to increased binding affinity to numerous cell surface receptors [40]. However, it remains unclear whether the synergistic action of MAA- and CIT- modified fibrinogen arises from binding to different receptors or whether co-modification alters protein conformation to enhance cell recognition efficiency. This question will be addressed in future studies. In addition, it is possible that other extracellular matrix proteins such as collagen and vimentin, alternative targets of ACPA and anti-MAA autoantibody [30,72], might also exhibit distinct patterns of macrophage activation and endothelial response. Future studies will be required to investigate these alternative protein substrates to assess broader patterns of macrophage activation and resulting endothelial responses in the context of RA-HF.

One-hour stimulation was chosen in this study to evaluate macrophage intracellular signaling, as this time-point was previously identified as optimal following FIB-MAA-CIT stimulation [37]. Additional time-points will be required in the future to fully characterize the signaling dynamics and assess the interaction between NF-κB and p38 pathways.

Additionally, we have evaluated responses in both U-937-derived and PBMC-derived macrophages, given the significant role of monocyte-derived macrophages in cardiac diseases [73,74]. However, cardiac-resident macrophages may have unique functions in myocardial inflammation and fibrosis and produce different cytokine responses following FIB-MAA-CIT stimulation, a possibility that warrants further investigation. Similarly, the endothelial cells used in this study were obtained from a single healthy donor. Prior studies have shown that effector cells, such as PBMCs and fibroblasts, that are derived from RA patients exhibit heightened responses to FIB-MAA-CIT stimulation compared to those from healthy controls [36,37]. Thus, our current findings may underestimate the inflammatory activation of endothelial cells in the context of chronic RA, thus making our results a conservative estimate of clinical realities.

In summary, our findings support a novel mechanism by which endogenously modified proteins drive macrophage–endothelial cell crosstalk, promoting myocardial inflammation. MAA and CIT adducts are increased and strongly co-localized in the myocardium and perivascular spaces of individuals with RA-associated HF. The potential relevance of this finding is underscored by our study showing that proteins co-modified with MAA and CIT directly activate HCAECs, promoting endothelial inflammation. In addition, co-modified fibrinogen synergistically activates macrophages through NF-κB and p38 pathways, resulting in the secretion of soluble mediators that further promote endothelial inflammation. Collectively, these findings suggest that therapeutic strategies targeting MAA, CIT, or their downstream effectors hold promise for mitigating cardiovascular complications in RA patients.

## Figures and Tables

**Figure 1 cimb-47-00943-f001:**
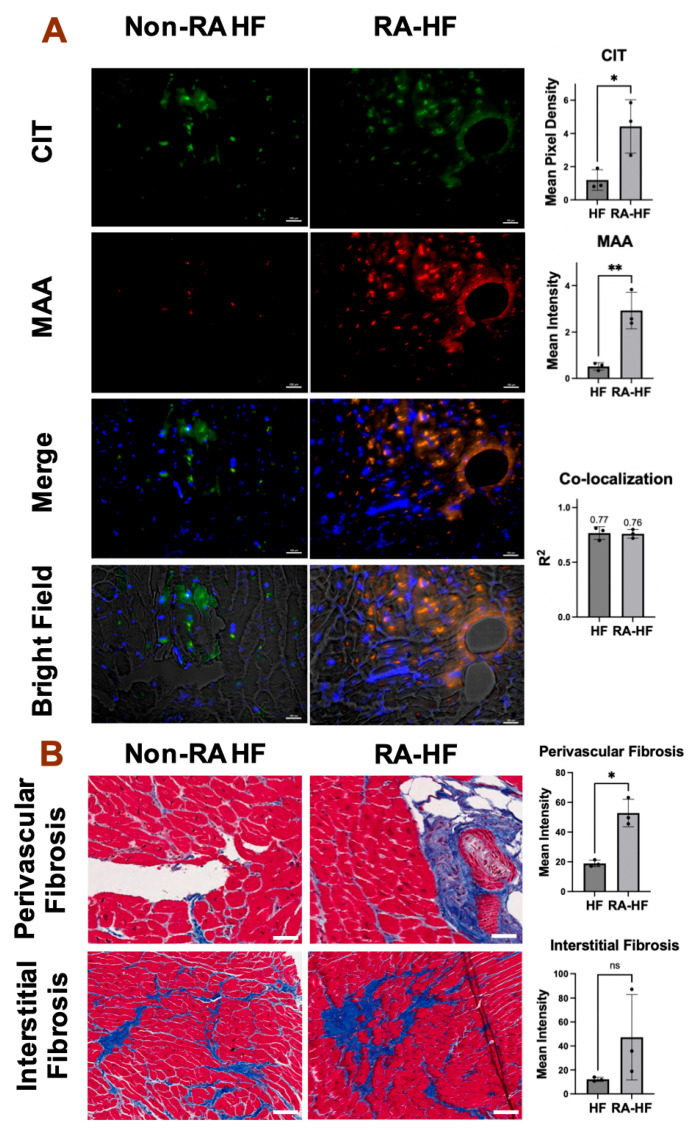
**Increased MAA expression, MAA-CIT co-localization, and perivascular fibrosis in RA-HF hearts.** Cardiac tissues from non-RA HF and RA-HF patients (n = 3 each) were sliced and imaged. Representative images of 20× frame at the apex of left ventricle are shown. (**A**) Immunohistochemistry using anti-CIT and anti-MAA antibodies. CIT (green) and MAA (red) expression and co-localization (orange) were quantified with the Fiji plug-in. scale bar = 100 µm. (**B**) Trichrome staining. Collagen deposition (blue) was quantified with the Fiji plug-in. Data shown in graphs are mean ± standard deviation. Student’s *t*-test was performed and statistical differences between groups are shown: * *p* < 0.05, ** *p* < 0.01. White scale bar = 100 µm.

**Figure 2 cimb-47-00943-f002:**
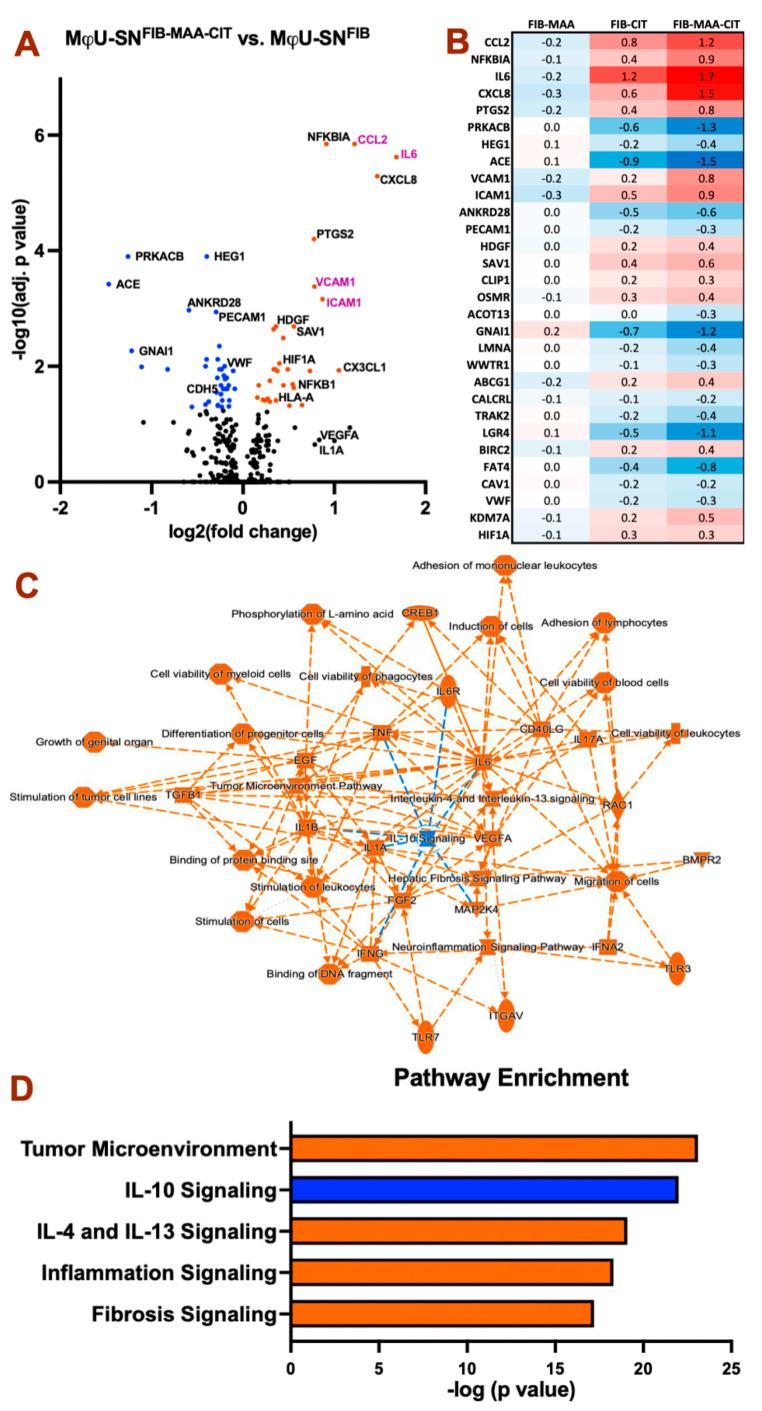
**HCAEC differential gene expression after treatment with antigen-stimulated macrophage supernatant.** Gene expression was compared with control HCAECs treated with MϕU-SN^FIB^ 8 h post-stimulation. (**A**) Volcano plot showing MϕU-SN^FIB-MAA-CIT^ vs. MϕU-SN^FIB^. Colored dots indicate significant changes (Adjusted *p* < 0.05). Genes in purple text are the top two inflammatory cytokines and leukocyte adhesion molecules. (**B**) Heat map showing log2 fold change in the top 30 differential expressed genes comparing MϕU-SN^FIB-MAA^, MϕU-SN^FIB-CIT^, and MϕU-SN^FIB-MAA-CIT^ to Mϕ-SN^FIB^. (**C**) Summary interaction graph between enriched genes and pathways. (**D**) Top enriched canonical pathways comparing MϕU-SN^FIB-MAA-CIT^ vs. MϕU-SN^FIB^. For panel (**A**–**D**), orange: positive z score (upregulation), blue: negative z score (downregulation).

**Figure 3 cimb-47-00943-f003:**
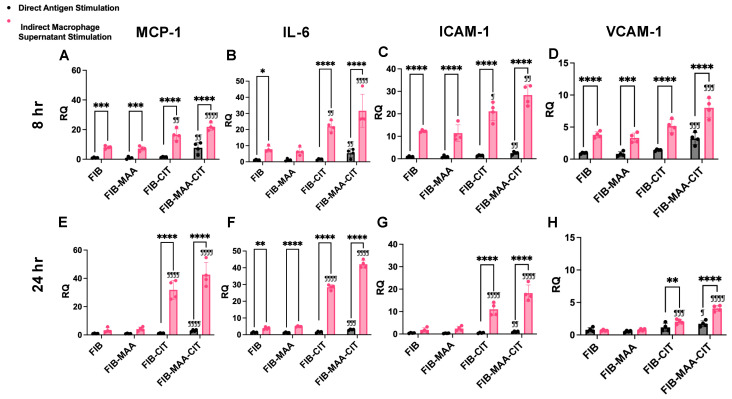
**HCAEC gene expression following direct antigen stimulation and indirect stimulation with macrophage supernatants.** MCP1, IL6, ICAM1, and VCAM1 mRNA expression was measured by RT-PCR in HCAEC 8 h (panel (**A**–**D**)) and 24 h (panel (**E**–**H**)) after stimulation. Relative quantification (RQ) compared to media-cultured negative control cells was calculated and is shown on the *y*-axis. Two-way ANOVA was performed, and the statistical differences between direct and indirect stimulation are shown: * *p* < 0.05; ** *p* < 0.01; *** *p* < 0.001; **** *p* < 0.0001. Statistical differences between CIT- and/or MAA-modified FIB compared with unmodified FIB are shown: ¶ *p* < 0.05; ¶¶ *p* < 0.01; ¶¶¶ *p* < 0.001; ¶¶¶¶ *p* < 0.0001.

**Figure 4 cimb-47-00943-f004:**
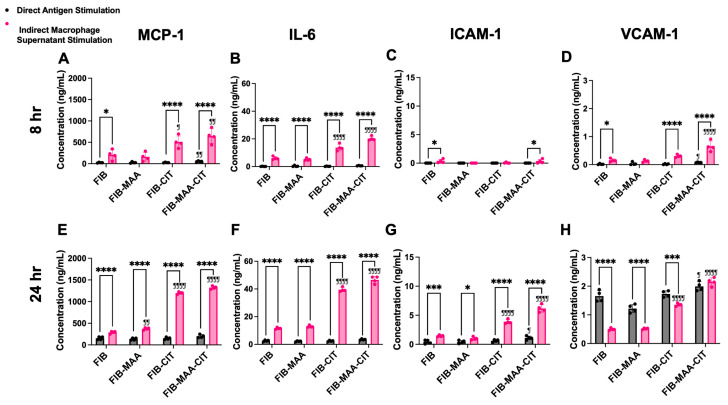
**HCAEC protein secretion following direct antigen stimulation and indirect stimulation with macrophage supernatants.** MCP-1, IL-6, ICAM-1, and VCAM-1 protein concentrations in HCAEC supernatants were measured by ELISA 8 h (panel (**A**–**D**)) and 24 h (panel (**E**–**H**)) after stimulation. Two-way ANOVA was performed and the statistical differences between direct and indirect stimulation are shown: * *p* < 0.05; *** *p* < 0.001; **** *p* < 0.0001. Statistical differences between CIT- and/or MAA-modified FIB compared with unmodified FIB are shown: ¶ *p* < 0.05; ¶¶ *p* < 0.01; ¶¶¶¶ *p* < 0.0001.

**Figure 5 cimb-47-00943-f005:**
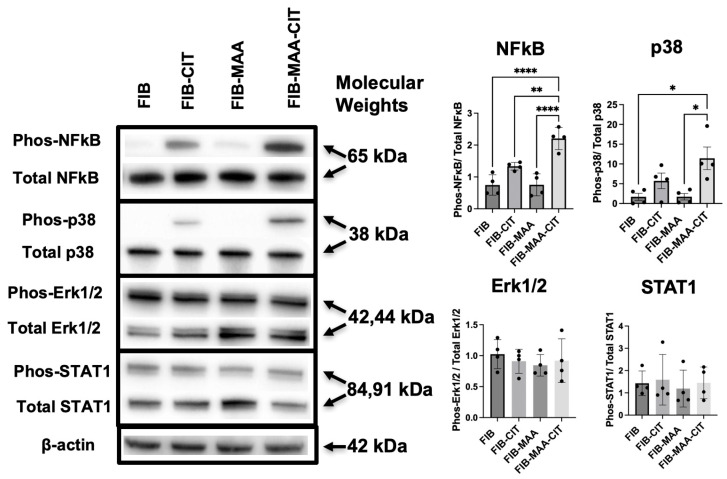
**FIB-MAA-CIT activates NF-κB and p38 pathways in macrophages.** Representative images of phosphorylated and total NF-κB, p38, Erk1/2, and STAT1 are shown, with β-actin as a loading control. Band intensities were quantified, and the ratio between phosphorylated and total protein was calculated and graphed. One-way ANOVA was performed and the statistical differences between groups are shown: * *p* < 0.05; ** *p* < 0.01; **** *p* < 0.0001.

**Figure 6 cimb-47-00943-f006:**
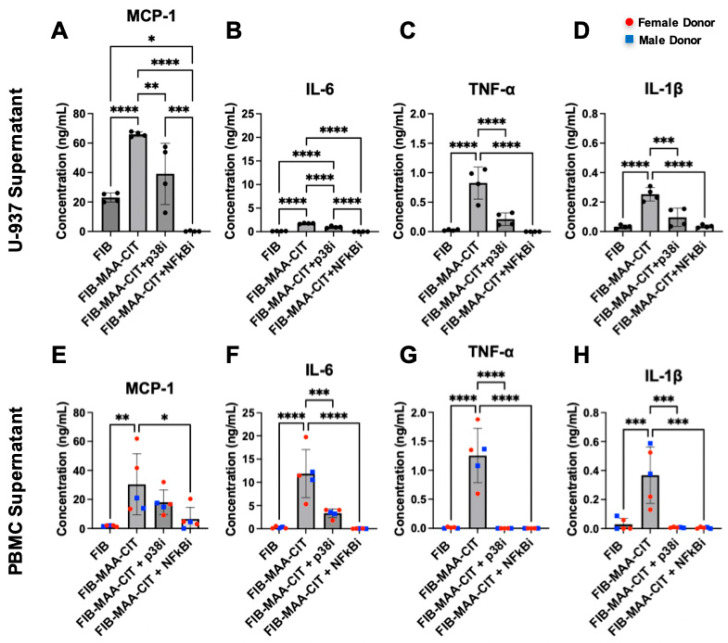
**Inhibition of p38 and NF-κB reduces macrophage cytokine production.** 48 h cytokine concentrations in MϕU-SN were measured by ELISA for (**A**) MCP-1, (**B**) IL-6, (**C**) TNF-α, and (**D**) IL-1β. 48 h cytokine concentrations in MϕP-SN were measured for (**E**) MCP-1, (**F**) IL-6, (**G**) TNF-α, and (**H**) IL-1β. One-way ANOVA was performed and the statistical differences between groups are shown: * *p* < 0.05; ** *p* < 0.01; *** *p* < 0.001; **** *p* < 0.0001. For panels (**E**–**H**), red circles represent female PBMC donors, and blue squares represent male PBMC donors.

**Figure 7 cimb-47-00943-f007:**
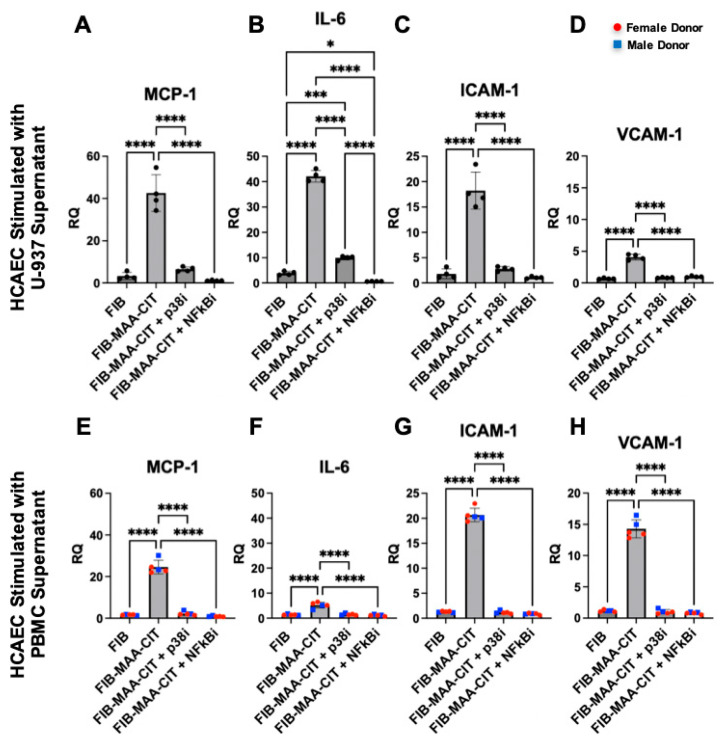
**HCAEC inflammatory gene expression following p38 and NF-κB inhibition of macrophages.** HCAEC mRNA expression after 24 h stimulation with MϕU-SN was measured by RT-PCR for (**A**) IL6, (**B**) MCP1, (**C**) ICAM1, and (**D**) VCAM1. HCAEC mRNA expression after 24 h stimulation with MϕP-SN was measured for (**E**) IL6, (**F**) MCP1, (**G**) ICAM1, and (**H**) VCAM1. Relative quantification (RQ) compared to media-cultured negative control cells was calculated and is shown on the *y*-axis. One-way ANOVA was performed and the statistical differences between groups are shown: * *p* < 0.05; *** *p* < 0.001; **** *p* < 0.0001. For panels (**E**–**H**), red circles represent female PBMC donors, and blue squares represent male PBMC donors.

**Figure 8 cimb-47-00943-f008:**
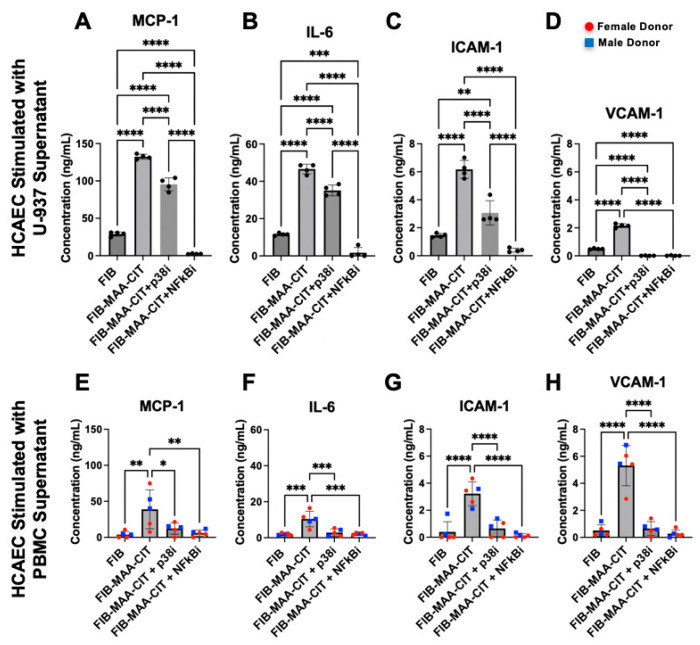
**HCAEC inflammatory protein release following p38 and NF-κB inhibition of macrophages.** HCAEC supernatant protein concentrations after 24 h stimulation with MϕU-SN were measured by ELISA for (**A**) IL-6, (**B**) MCP-1, (**C**) ICAM-1, and (**D**) VCAM-1. HCAEC supernatant protein concentrations after 24 h stimulation with MϕP-SN were measured for (**E**) IL-6, (**F**) MCP-1, (**G**) ICAM-1, and (**H**) VCAM-1. One-way ANOVA was performed and the statistical differences between groups are shown: * *p* < 0.05, ** *p* < 0.01; *** *p* < 0.001; **** *p* < 0.0001. For panels (**E**–**H**), red circles represent female PBMC donors, and blue squares represent male PBMC donors.

## Data Availability

The original contributions presented in this study are included in the article/Appendix A. Further inquiries can be directed to the corresponding author.

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
