# Peer review of "Citrullinated and Malondialdehyde–Acetaldehyde-Modified Fibrinogen Activates Macrophages and Promotes Coronary Endothelial Cell Inflammation"

_cimb, 2025, doi:10.3390/cimb47110943_

Round 1
Reviewer 1 Report
Comments and Suggestions for Authors
Dear Author,
This manuscript addresses an important question linking protein post-translational modifications (CIT and MAA) to macrophage activation and endothelial inflammation in RA-associated heart failure. The experimental design is strong, and the study includes several sophisticated approaches that provide new insights. However, the writing and organization limit the impact of the findings. Below are specific points that could help strengthen the manuscript:
Strengths
• The methodological design is thoughtful. Comparing multiple time points (8 h, 24 h, 7 days) adds depth, showing both early transcriptional events and later protein-level or sustained responses. This helps build a more complete picture of how FIB-MAA-CIT drives inflammation.
• The combination of macrophage and endothelial cell assays is valuable for modeling crosstalk relevant to RA-HF.
Points for Improvement
1. Writing clarity and interpretation
o Several sentences are unnecessarily complex and difficult to follow. For example, the first sentence started at line 130 and finished at line 134, contained 54 words, which makes it super long. The second example, the description of p38 phosphorylation results (line ~230) tries to pack multiple comparisons into one sentence, which confuses the logic. Please simplify and separate comparisons clearly.
o Line ~267 is also confusing and should be rewritten for clarity. There are plenty of such sentences. I just pointed a few.
2. Overstatements
o Line 129: The statement “FIB-MAA-CIT Promotes Coronary Endothelial Inflammation via Macrophage Interaction” is too strong given the data. The experiments model macrophage–endothelial crosstalk in vitro, but do not establish this mechanism as a definitive driver in vivo. Please moderate the wording.
3. Cell models
o Line 135: U-937 cells are a useful macrophage model, but they have limitations. They do not fully capture the diversity of tissue-resident macrophages, and cytokine responses may differ from primary MDMs or cardiac macrophages. This limitation should be more explicitly acknowledged.
4. Experiment rationale
o Line 242: The NF-κB and p38 inhibition experiments are important, but the rationale should be better explained. For example: “to test whether NF-κB and p38 activation are functionally required for FIB-MAA-CIT–induced cytokine secretion and to dissect pathway specificity.” Simply showing phosphorylation does not prove causality. Furthermore, author screened 9 pathways, but most were “undetectable.” That could mean either no activation or limitations of assay sensitivity/timing. For instance, STATs and Akt often have transient phosphorylation peaks at earlier time points (<30 min). Testing only a 1-hour timepoint risks missing early or late signaling dynamics.
o Additionally, although both NF-κB and p38 are shown to be involved, this information was already established in the authors’ prior publication [DOI: 10.1152/ajplung.00153.2024]. What would add more value here is clarification of upstream–downstream relationships. Do NF-κB and p38 act in parallel, or is one upstream of the other? The inhibitor data confirm dependency but do not prove causality or exclusivity.
o Also, in Figure 6, both sexes (male and female) were used, but this is never acknowledged or discussed in the text. Please explain why this is relevant.
5. Figures
o There is missing description of Figure 7E–H in the text.
6. Discussion structure
o The discussion spends too much time summarizing RA/HF inflammation literature (refs 32–47) and therapeutic agents (refs 52–63). This reads like an introduction or literature review. Instead, please focus first on your own results, then place them in the context of prior work.
o Interesting findings, such as downregulation of PECAM1/VWF and upregulation of fibrosis pathways, suggesting endothelial–mesenchymal transition, however this information is buried within dense reference-heavy text. These should be highlighted as novel hypotheses arising directly from author’s data.
o The rationale for studying fibrinogen, which is central to the study, is well described in the be introduced however, it should be reiterated in the beginning of the discussion, not buried near the end.
7. Limitations
Limitations
• The limitations should be presented in a clearly defined subsection. Important points include: (a) small sample size (n=3), which is underpowered and highly sensitive to outliers and heterogeneity; (b) lack of adjudication of heart failure subtype (HFrEF vs HFpEF, ischemic vs non-ischemic, etc.); (c) no information on RA duration, activity, ACPA/RF serostatus, or immunomodulatory therapy (methotrexate, TNF/IL-6 inhibitors, steroids), all of which can influence citrullination and MAA formation.
• Notably, although all samples had reduced EF (suggesting HFrEF), HFpEF is common in RA and could confound RA-specific inferences.
Study design concern
• The overall study design appears highly similar to the authors’ two previous publications [DOI: 10.1152/ajplung.00153.2024; DOI: 10.3389/fimmu.2023.1203548]. This raises my strong concerns about redundancy and whether sufficient novelty is demonstrated. The authors should make clearer how the present work advances beyond these prior studies. I understand that author presented different cell types but the originality is very low.
Author Response
RESPONSE 1: We thank the reviewer for the kind remarks.
RESPONSE 2: We thank the reviewer for the valuable feedback on the need for improved clarity to better facilitate interpretation. We have revised our manuscript to simplify long sentences and to present comparisons more clearly throughout.
RESPONSE 3: We agree with the reviewer that FIB-MAA-CIT mediated endothelial-macrophage crosstalk has not been adequately established in vivo to support this statement. Therefore, we have modified the header for Section 2.2 to more appropriately read: “FIB-MAA-CIT Promotes Coronary Endothelial Inflammation in-vitro via Macrophage Interaction” (See lines 135-136 of the marked up manuscript). We have similarly modified language in the abstract to clarify that crosstalk was established in vitro (See line 33 of the marked up manuscript).
RESPONSE 4: We agree with the reviewer that cardiac tissue-resident macrophages may respond differently than monocyte-derived macrophages used in our experiments. In response to this comment, we have expanded our discussion section to address this highly relevant limitation.
To the discussion, we have added: “However, cardiac-resident macrophages may have unique functions in myocardial inflammation and fibrosis and produce different cytokine responses following FIB-MAA-CIT stimulation, a possibility that warrants further investigation.” (See lines 454-456 of the marked up manuscript).
RESPONSE 5: We appreciate the reviewer’s suggestion to bolster the rationale for experiments inhibiting NF-kB and p38. In response to this comment, we have modified the methods section to include the wording kindly suggested by the reviewer: “To test whether NF-κB and p38 activation are functionally required for FIB-MAA-CIT–induced cytokine secretion and to dissect pathway specificity, inhibitors of these pathways were utilized.” (See lines 654-655 of the marked up manuscript).
The 1-hour time point was selected as it was previously demonstrated to be optimal for detecting intracellular signaling following FIB-MAA-CIT stimulation (see PMID: 37654483). We agree that additional timepoints will be required in the future to more fully characterize the signaling dynamics and assess for possible interactions between NF-κB and p38 pathways. This has been added as a limitation in our discussion section: “One-hour stimulation was chosen in this study to evaluate macrophage intra-cellular signaling, as this time-point was previously identified as optimal following FIB-MAA-CIT stimulation.[27] Additional time-points will be required in the future to fully characterize the signaling dynamics and assess the interaction between NF-κB and p38 pathways.” (See lines 447-451 of the marked up manuscript)
RESPONSE 6: We appreciate the opportunity to clarify the novelty of NF-κB and p38 findings in our study. Although we have previously examined p38 pathways in response to FIB-MAA-CIT stimulation, p38 was not detected in the cell types studied (joint and lung fibroblasts; PMID: 37654483, 39560968). This is the first paper that shows FIB-MAA-CIT activates both NF-κB and p38 in macrophages. The novelty of this finding is now clarified in the discussion: “In addition, this is the first report to examine FIB-MAA-CIT activation of any cardiac cell types and the first to reveal NF-κB and p38 as novel pathways activated in response to FIB-MAA-CIT stimulation of macrophages.” (See lines 310-313 of the marked up manuscript).
We also agree with the reviewer that clarifying upstream-downstream relationships between NF-κB and p38 pathways is important, and we plan to examine this in future studies. This important point is now acknowledged in our expanded discussion: “Additional time-points will be required in the future to fully characterize the signaling dynamics and assess the interaction between NF-κB and p38 pathways”. (See lines 449-451 of the marked up manuscript).
RESPONSE 7: We thank the reviewer for pointing this out. We have added additional information regarding to our methods section: “Both male and female donors were included to enhance the generalizability of our results.” (See lines 495-496 of the marked up manuscript).
RESPONSE 8: Figure 7E-H is described at the end of section 2.4: “These findings were confirmed using MÏ•P-SNFIB-MAA-CIT (PBMC-derived macrophages from 5 separate human donors), showing similar responses (Fig. 7E-H,8E-H).” (See line 287 of the marked up manuscript).
RESPONSE 9: We thank the reviewer for the suggestions on the structure of the discussion. In response to this comment, we have rearranged the discussion section by first summarizing our results (including the novel findings noted above that highlight EndoMT as a hypothesis generating observation) and then contextualizing these findings with a more streamlined/targeted review of prior work.
RESPONSE 10: We thank the reviewer for this comment. We have now reiterated the rationale for studying fibrinogen in the beginning of the discussion section. (See lines 322-324 of the marked up manuscript).
RESPONSE 11: We agree that the limited sample size and the lack of information on HF and RA disease characteristics limit the generalizability of this study. We incorporated reviewer’s verbiage into our discussion as a clearly defined limitation subsection. (See section 3.1, lines 402-405 of the marked up manuscript).
RESPONSE 12: We appreciate the opportunity to clarify the originality and novelty of the work presented. As noted by the reviewer, our group has pioneered a novel model to examine in vitro effects of dual protein modifications in RA. To our knowledge, these prior reports and the current report are among the only studies completed to date leveraging this approach. It is perhaps then not surprising that herein, we have leveraged this unique model to study a highly relevant RA-related complication for the very first time. To our knowledge, this report is the first to report increased expression of MAA adducts in RA myocardium, strong MAA-CIT co-localization, and increased perivascular fibrosis in heart tissue of RA patients with heart failure compared to non-RA controls. In addition, this is the first report to examine FIB-MAA-CIT activation of any cardiac cell types and the first to reveal NF-κB and p38 as novel pathways activated in response to FIB-MAA-CIT stimulation. Overall, we believe this paper provides novel mechanistic insights into how endogenously modified proteins may contribute to myocardial inflammation in RA and is suitable for publication in IJMS.
In response to this comment and comment #9 above, we have modified the discussion to better articulate both the originality and novelty of this study.
Reviewer 2 Report
Comments and Suggestions for Authors
Thank you for the opportunity to review this study links two rheumatoid-arthritis–relevant post-translational modifications—citrullination (CIT) and malondialdehyde-acetaldehyde (MAA) adduction—to myocardial inflammation in RA-associated heart failure (RA-HF).
Human myocardium data are linked to cell-based macrophage–endothelial crosstalk experiments with coherent readouts (gene, protein, signaling).
Co-modified fibrinogen (MAA+CIT), but not single modifications, drove the strongest responses; indirect (macrophage-conditioned) stimulation exceeded direct antigen exposure.
NF-κB and p38 activation in macrophages was shown, and pathway inhibition attenuated both macrophage cytokines and downstream endothelial activation.
Several important issues should be noted.
Major comments:
1. Small, under-characterized human tissue cohort. Only 3 RA-HF and 3 non-RA HF samples. Expand the myocardial cohort with clinical phenotyping (including HFpEF), and analyze PTM signal in relation to RA features and therapies.
2. Macrophages were U-937–derived or blood-derived; cardiac-resident macrophages were not studied.
3. Antigen dose (25 µg/mL) and degree/order of modification were not related to in vivo myocardial levels; residual antigen carry-over in macrophage supernatants was not quantified. Functional endothelial outcomes (adhesion under flow, barrier integrity, EndoMT markers) were not shown despite pathway hints.
The author can consider integrate these into their study.
Addressing these limitations should improve this manuscript. I wish you continued success in this work. I recommend "major revision" for this manuscript.
Author Response
- Small, under-characterized human tissue cohort. Only 3 RA-HF and 3 non-RA HF samples. Expand the myocardial cohort with clinical phenotyping (including HFpEF), and analyze PTM signal in relation to RA features and therapies.
RESPONSE: We agree with the reviewer that the availability of only 3 myocardial samples from patients with both RA and heart failure renders our results prone to outliers and may limit generalizability of our findings. Though we have access to a large cardiovascular biobank that includes 880 myocardial tissue samples, we were only able to identify 3 patients with definite RA who had undergone either LVAD placement or transplant. We used all available samples for this study. In addition to the highly invasive nature of cardiac biopsy, coupled with a low disease prevalence of RA (0.5% worldwide), we believe our limited tissue access is also impacted by the fact that RA patients suffer from more severe HF, rendering these patients less likely to receive either LVAD or transplant. If the reviewer and/or editors are aware of other cardiac biobanks that could serve as a resource, we would be delighted to extend our studies further.
To fully address this comment, we have expanded our limitation section on page 17-18:
“There are limitations to this study. The small number of cardiac tissues available for this study renders it sensitive to outliers and individual heterogeneity. In addition, the lack of adjudication of heart failure subtype and RA disease characteristics could limit the RA-specific inferences from these observations. Notably, all RA-HF patients and non-RA HF controls had advanced heart failure with ejection fractions below 35% at the time of tissue collection. Moreover, information specific to the underlying etiology of myocardial dysfunction was unknown…. Although the limited number of samples examined renders our results susceptible to outliers and may limit generalizability to larger populations, these data are consistent with preliminary observations in collagen-induced arthritis mice [ACR abstract ID 0065] in addition to prior human studies showing increased citrullination in RA myocardium and the presence of MAA in coronary atheromas [PMID 22364592, 25210746]”
- Macrophages were U-937–derived or blood-derived; cardiac-resident macrophages were not studied.
RESPONSE: We utilized monocyte-derived macrophages as these represent a key cell type involved in promoting myocardial inflammation and heart failure (PMID:29301844, 38816371). We agree that cardiac-resident macrophages may have a unique response pattern to modified-proteins, and this will be investigated in future studies. In response to the current and prior reviewer’s comment, we have included the follow paragraph in our limitation section:
“Additionally, we have evaluated responses in both U-937-derived and PBMC-derived macrophages, given the significant role of monocyte-derived macrophages in cardiac diseases [ PMID:29301844, 38816371]. However, cardiac-resident macrophages may have unique functions in myocardial inflammation and fibrosis and produce different cytokine responses following FIB-MAA-CIT stimulation, a possibility that warrants further investigation.” (page 19)
- Antigen dose (25 µg/mL) and degree/order of modification were not related to in vivo myocardial levels;
RESPONSE: The 25 µg/mL dose of modified fibrinogen was used because it was previously shown as the optimal concentration to activate macrophages in vitro (PMID: 35785731, 37654483, 39560968). While the physiological concentration of fibrinogen in heart tissue is unknown, the 25 µg/mL used in this study is approximately 1% of the fibrinogen concentration found in normal human plasma (PMID: 35008616). This concentration reflects a conservative estimate of perivascular fibrinogen concentration during chronic inflammatory conditions such as RA and hear failure (PMID: 12146723, 40229608, 38731063).
We agree with the reviewer that no study, to our knowledge, has investigated the degree of MAA- and/or CIT-modified fibrinogen in myocardial tissues. To adequately address the reviewer’s comment, we have expanded our discussion section to address this limitation:
“The 25 µg/mL dose of modified fibrinogen was used for cell stimulations because it has been previously shown as the optimal concentration to activate macrophages in our in vitro model (PMID: 35785731, 37654483, 39560968). However, whether this concentration reflects the in vivo concentration of modified fibrinogen in myocardial tissue is unknown and warrants future investigation.” (page 18)
- Residual antigen carry-over in macrophage supernatants was not quantified.
RESPONSE: The reviewer raises a valid point. While residual antigen carry-over in macrophage supernatants was not quantified, we designed our experiment in a way so that we could estimate the effect of any carry-over antigens. We stimulated endothelial cells both directly with antigens and indirectly with macrophage supernatants (that contains both macrophage secretions and carry-over antigens). The effect of modified antigen on endothelial cells is shown through direct stimulation, and the effect of macrophage secretions on endothelial cell can be estimated by subtracting the effect of direct stimulation from that of indirect stimulation.
To make this point more clearly, we have included following in our Methods section:
“To account for the potential influence of residual FIB antigens in MÏ•U-SN or MÏ•P-SN, HCAECs were also directly stimulated with 25 μg/mL of unmodified FIB, FIB-MAA, FIB-CIT, or FIB-MAA-CIT for comparison in separate experiments (Fig. S1).” (page 5)
We have also expanded our limitations section to include:
“In addition, the residual antigen carry-over in macrophage supernatants was not quantified. To account for the potential influence of residual antigens in macrophage supernatants, endothelial cells were also directly stimulated with modified antigens for comparison in separate experiments.” (page 18)
- Functional endothelial outcomes (adhesion under flow, barrier integrity, EndoMT markers) were not shown despite pathway hints.
RESPONSE: We thank the reviewer for this suggestion. This manuscript focused mechanisms behind PTM-induced endothelial inflammation, because inflammation was the predominant response revealed by the NanoString Cardiovascular Panel. We agree with the Reviewer, however, that these findings are important hypothesis-generating observations that will require additional experiments to elucidate. As such, we are conducting ongoing experiments that will comprehensively assess EndMT markers and endothelial cell morphology and function. These needed experiments are beyond the scope of the report under review and we anticipate that these results will be presented in a subsequent report.
To address the point raised by the Reviewer, we have included the following paragraph in our Discussion section:
“This study primarily focused on inflammatory endothelial cell responses, as this was the predominant response that we observed 8 hours after indirect stimulation with MÏ•U-SNFIB-MAA-CIT. However, our NanoString data also showed downregulation of endothelial markers such as PECAM1 and VWF, alongside upregulation of the tumor microenvironment and fibrosis-related pathways in HCAECs. These data support a novel hypothesis that MÏ•U-SNFIB-MAA-CIT-stimulated endothelial cells may begin to lose endothelial characteristics and start to acquire a pro-fibrotic, mesenchymal phenotype. Previous studies have shown that chronic stimulation of endothelial cells with IL-1β, TNF-α, or TGF-β can induce transformation to a mesenchymal phenotype.[64–66] Since our current and previous study[26] have shown that co-modified fibrinogen increases macrophage secretion of IL-1β, TNF-α, and TGF-β, chronic indirect stimulation with MÏ•-SNFIB-MAA-CIT could lead to a robust pro-fibrotic phenotype in endothelial cells, further contributing to perivascular fibrosis in RA. Future studies will be needed to characterize this pro-fibrotic endothelial phenotype more precisely and to identify the core mediators of this macrophage-endothelial crosstalk.” (Page 18)
Round 2
Reviewer 1 Report
Comments and Suggestions for Authors
The authors have addressed the previously raised concerns in a satisfactory manner. The revisions made have substantially improved the clarity and overall quality of the manuscript.
Author Response
- The authors have addressed the previously raised concerns in a satisfactory manner. The revisions made have substantially improved the clarity and overall quality of the manuscript.
RESPONSE: We thank the reviewer their positive comments regarding our revised manuscript and our responsiveness to previous critiques.
Reviewer 2 Report
Comments and Suggestions for Authors
The authors have addressed all the questions.